# Association of Hypertension, Diabetes, and Cardiovascular Disease with COVID-19 in Africa: Scoping Review Protocol

**DOI:** 10.3390/tropicalmed8060293

**Published:** 2023-05-26

**Authors:** Faisal Nooh, Jürg Utzinger, Daniel H. Paris, Nicole Probst-Hensch, Afona Chernet

**Affiliations:** 1Swiss Tropical and Public Health Institute, Kreuzstrasse 2, CH-4123 Allschwil, Switzerland; 2University of Basel, CH-4003 Basel, Switzerland; 3College of Medicine & Health Sciences, University of Hargeisa, Hargeisa, Somalia; 4College of Medicine & Health Sciences, Jigjiga University, 1020 Jigjiga, Ethiopia

**Keywords:** Africa, cardiovascular diseases, COVID-19, diabetes, hypertension, noncommunicable diseases, pandemic, SARS-CoV-2, severity

## Abstract

Background: COVID-19 caused devastating effects on global healthcare systems. The elderly and people with chronic comorbidities were at a particularly high risk of mortality and morbidity. However, the evidence on the association of COVID-19 severity with noncommunicable diseases (NCDs) in the African population is scarce. Objective: The aim is to estimate COVID-19 severity among African patients with hypertension, diabetes, and cardiovascular diseases (CVDs) and its implications for case management. Methods: We will adhere to the extension for Scoping Reviews of PRISMA (PRISMA-ScR). The following electronic databases will be searched: PubMed, Scopus, Web of Science, Embase, CINAHL, and Joanna Briggs Institute. The search will be conducted after the publication of this protocol. Two reviewers will extract data from articles published after March 2020 without language restrictions. A descriptive analysis of the important findings and a narrative synthesis of the results will serve as the basis for interpretation. Expected results and conclusions: This scoping review is expected to determine the odds of patients with chronic comorbidities to progress to severe stages of COVID-19. The review will generate an evidence-based and set foundation for recommendations toward the establishment of surveillance systems and referral guidelines for the management of NCDs in the face of COVID-19 and future pandemics.

## 1. Introduction

Coronavirus disease 2019 (COVID-19), is an illness caused by severe acute respiratory syndrome coronavirus 2 (SARS-CoV-2) [1]. The unprecedented spread of this infectious disease (pandemic) posed devastating effects on the health and well-being of people around the world, including Africa. As of 1 January 2023, according to the Africa Centre for Diseases Control and Prevention (Africa CDC) [2,3], there were more than 12.2 million confirmed cases and 256,542 deaths reported throughout Africa, representing about 2% of all cases (656.4 million) and about 4% of all deaths (6.7 million) reported globally.

The effect of the COVID-19 pandemic on global healthcare systems has been profound. In particular, the impact of the pandemic on the elderly and people with non-communicable diseases (NCDs) has been devastating [4,5,6,7,8]. Important issues include disordered regular service delivery, decreased access to existing healthcare facilities, social isolation, and supply chain disruptions [4,5,9]. Additionally, the COVID-19 pandemic severely affected access to and the utilization of healthcare facilities, management of chronic diseases, maternal and child services, vaccination programs and regular control, and treatment of endemic diseases, such as malaria, tuberculosis, HIV/AIDS, and neglected tropical diseases [6,10,11,12,13,14]. Interventions and clinical trials have also been adversely affected.

The increasing burden of NCDs along with the enduring burden of infectious diseases in Africa and other settings in low- and middle-income countries (LMICs) has been well noted [15]. Nevertheless, the treatment and control of NCDs have been routinely given less attention, as often priority is bestowed to the control and management of infectious diseases. Additionally, regular check-ups and early screenings are yet not well adapted. Treatment delays and low utilization rates of the available services have been among the daunting challenges in Africa and elsewhere in LMICs leading to a high prevalence of advanced NCD conditions and an increased burden of preventable complications. Moreover, inadequate self-management practices and nonadherence to treatment procedures of these lifelong conditions have been major hurdles to patients with NCDs. Given this existing dual burden of disease in Africa and the global impact of COVID-19, the pandemic presumably aggravated the already existing healthcare crisis in the continent [6,7,8,16,17].

Nonetheless, information on the effect of COVID-19 in Africa focused on health system challenges, and evidence on the effect of COVID-19 in African patients with hypertension, diabetes, and cardiovascular diseases (CVDs) was extrapolated from what was obtained in other parts of the globe. In order to properly allocate scarce resources and to support clinical decisions during the COVID-19 pandemic, it is important to have an evidence-based record, derived from available epidemiological and clinical data on the comorbidity between NCDs and COVID-19 among African patients. This is of particular importance as in many African countries, the demography is characterized by a large proportion of young people and a lower prevalence of lifestyle risks (e.g., obesity and smoking), which may prevent against severe SARS-CoV-2 [18,19,20,21]. Furthermore, genetic differences in COVID-19 susceptibility may exist [22]. To our knowledge, no study or report is available to date that summarizes the severity of COVID-19 in African patients with NCDs.

A preliminary search on International Prospective Register of Systematic Reviews (PROSPERO) [23], Open Science Framework (OSF) [24], and Joanna Briggs Institute (JBI) [25] showed that no scoping review on the association between COVID-19 and hypertension, diabetes, and CVDs is currently ongoing or registered. Hence, this scoping review is an attempt to fill this gap and generate evidence for improved management of COVID-19 and the aforementioned NCDs. Furthermore, it will inform African policy makers and healthcare professionals toward the establishment of surveillance systems and referral guidelines for the management of NCDs during the COVID-19 pandemic and future pandemics.

## 2. Aim and Review Questions

The overarching aim of this review is to focus on the potential factors related to the severity of COVID-19 for African patients with hypertension, diabetes, and CVDs and their implications for case management. The following research questions will guide to conduct this scoping review:What type of severity outcomes were reported in the included studies?What relative impact did the selected NCDs have on the severity of COVID-19?Are there specific patient characteristics that increase the risk of COVID-19 severity among patients with the selected NCDs, namely hypertension, diabetes, and CVDs?What strategies and interventions have addressed the risk factors for COVID-19 severity in comorbid patients?Which of the three selected NCDs had a major impact on exacerbating COVID-19 severity in Africa?What impact did the COVID-19 response have on the services for NCDs in Africa?

## 3. Materials and Methods

### 3.1. Protocol and Registration

The scoping review will be conducted in accordance with the guidance for pursuing systematic scoping reviews, put forth by Peters and colleagues from JBI in Australia [26]. The methodologies comply with the extension for Scoping Reviews of the Preferred Reporting Items for Systematic Reviews and Meta-Analyses (PRISMA-ScR), given in Appendix A [27]. The review protocol was registered on OSF (registration link: http//:osf.io/e9r28 accessed on 12 May 2023) on 17 November 2022.

### 3.2. Inclusion Criteria

Studies focusing on COVID-19 patients meeting the following criteria will be included: (i) studies that estimated the quantitative relationship between COVID-19 and hypertension, diabetes, and CVDs; (ii) studies conducted on the African continent; (iii) and studies of both observational (longitudinal and cross-sectional) and interventional (randomized and nonrandomized community trials and controlled and uncontrolled before/after studies) designs.

### 3.3. Exclusion Criteria

Not considered will be studies that met at least one of the following exclusion criteria: (i) studies that evaluate COVID-19 patients without considering NCDs and vice versa; (ii) position papers, editorials, policy statements, case reports, case series studies, perspectives, commentaries, published abstracts, poster and oral presentations, and author reply articles; (iii) studies on the potential association between COVID-19 and other infectious diseases, malignancies, or autoimmune disorders, but not considering diabetes and hypertension; and (iv) articles that speculatively extrapolate findings from studies conducted outside Africa to explain the effects of COVID-19 in Africa.

### 3.4. Participants

The literature search will include results from studies reporting on COVID-19 and NCD comorbidities among African patients. The review will include studies on adults aged ≥18 years, irrespective of their gender. Data on patients participating in clinical trials, cross-sectional epidemiological studies, or cohort studies (both retrospective and prospective) will be included in the review. Data on patients admitted to any healthcare facility, including outpatient departments, emergency rooms, and intensive care units (ICUs) and reporting to have comorbidities of COVID-19 and NCDs are eligible for the review. However, the review will not consider studies that report from African diaspora patients (living outside of Africa).

### 3.5. Concept

The review will address the severity of COVID-19 symptoms among African patients due to either one or several of the selected NCDs. For pragmatism and homogeneousness, the review will consider the standard definitions of COVID-19 severities set forth by the World Health Organization (WHO). The common categories for the level of aggravation of COVID-19 among adult population are nonsevere (mild or early stage), moderate, severe, and critical [28]. Moreover, as many studies from our preliminary search mentioned “asymptomatic” as a category in their findings, we will include it in the search outcome, in addition to the categories used by WHO [29]. The review will uncover the weight of multicomorbidity of NCDs in further aggravating COVID-19 infection, changing the pattern or course of outcomes. Patient characteristics, which weigh in exacerbating the COVID-19 condition, will be marked. The final analysis will map and elaborate the morbidity outcome in relation to the major NCDs among African patients and identify strategies and intervention management for NCD comorbidity during pandemics on the continent.

### 3.6. Information Sources and Search Strategy

The systematic search strategy will mainly be aimed at published peer-reviewed articles. To identify potentially suitable articles, we will search documents from the following electronic databases: PubMed/MEDLINE, Embase/Elsevier, Scopus, Cumulative Index to Nursing & Allied Health Literature (CINAHL)/EBSCO, and Web of Science. Because the first COVID-19 case was confirmed in Africa on 14 February 2020, we will search the databases from March 2020 to 28 February 2023 (3-year period) without language restrictions.

A three-step search strategy will be used in this review. First, an initial search of PubMed will be undertaken followed by an analysis of the text words contained in the title and abstract and of the index terms used to describe the article. Second, using all the identified keywords and index terms, we will search all the other databases. Third, we will undertake a hand search of the reference list of all the identified relevant documents for potential additional articles. An example of the terms and strings of words applied on PubMed is provided in Appendix B.

### 3.7. Search Results

Records retrieved through the aforementioned search strategy from all databases will be imported into the bibliometric software EndNote™ X9 (Clarivate Analytics; Philadelphia, PA, USA) and screened for relevance and duplication. The criteria for relevance are based on the scope and objectives of the review. Using the inclusion criteria set above, two reviewers will conduct full assessment of the identified scientific publications, and any duplicates will be removed. Any disagreement will be resolved through discussion, and with a third reviewer, as the case might be.

### 3.8. Data Charting Process

The reviewers will develop a data abstraction tool to capture relevant information from the selected documents. The tool will encompass detailed information that includes (i) participant characteristics, such as demographics of patients; (ii) study characteristics, such as study setting, study types, publication dates, authors, methodology, etc.; and (iii) outcomes and key findings related to the review objective. Two reviewers will independently chart information from each selected document to ensure charting consistency and inter-reviewer reliability. In case of disagreements, the two reviewers will resolve through discussion or in consultation with a third team member.

All extracted data will be exported to Microsoft^®^ Excel 2016 (Microsoft; Redmond, WA, USA). A draft data abstraction tool is provided in Appendix C. As the reviewers familiarize themselves with the content of the selected documents, necessary modifications to and revisions of the data abstraction tool will be made. We will include the final version of the data abstraction tool in the scoping review publication.

### 3.9. Data Analysis and Presentation

Outcome of the systematic search will be analyzed descriptively using frequencies and percentages. Moreover, graphical presentation including tables and charts will be used, whenever applicable. This will compare and/or reflect the effect of NCDs on the severity of COVID-19. Effect of comorbidities of selected NCDs (diabetes, hypertension, and CVDs) will be mapped accordingly and will be presented in comparison to those without any comorbidities.

Figure 1 illustrates the graphical summary of the methodological strategy applied for the protocol of the scoping review.

## 4. Results

In this section, we will summarize the results of the search strategy and the process of document selection, inclusion, and exclusion in both text and chart formats. We will tabulate and describe detailed information about the selected studies. Emphasis will be placed on the following information: authors, year of publication, country, aim of the study, study design, study setting, participant characteristics, sample size, main findings, measures of outcomes, and effect size (if relevant).

The abstracted information in relation to the objectives of the review will be summarized and presented in detail. For example, we will present the number of studies that examined associations between COVID-19 and hypertension. We will describe the relative frequencies of the studies by geographical location and number and characteristics of the participants included in terms of age, sex, and severity of COVID-19. Moreover, the types of study designs and outcome measurements will be described. Additionally, the types and strengths of the reported relationships between COVID-19 and hypertension will be given, and the consistency of the findings will be reported. We will also recount the effect of the COVID-19 response measures (e.g., lockdowns, closure of outpatient clinics, stock-outs of medicines, diagnostics, personal protective equipment, etc.) on patients.

After describing the reported relationship between COVID-19 and the three selected NCDs, we will report the relationship between COVID-19 and concurrent NCDs. The final synthesis will present the overall severity of COVID-19 in people with all the selected NCDs.

## 5. Conclusions

This scoping review is expected to determine the odds of patients with chronic comorbidities to advance into severe stages of COVID-19 and to estimate the extent to which NCD services were affected during the COVID-19 response in Africa. In doing so, the review will provide new evidence and set foundations for recommendations toward the establishment of surveillance systems and referral guidelines for the management of NCDs during the COVID-19 pandemic and future pandemics. The outcome of the review will increase awareness of healthcare professionals, policy makers, and other key stakeholders. It will also enhance further collaboration among research, surveillance systems, and technological advancement to develop new advanced diagnostic tools and to set up policies that deal with the silent pandemic of the dual burden of NCDs and infectious diseases in Africa.

## Figures and Tables

**Figure 1 tropicalmed-08-00293-f001:**
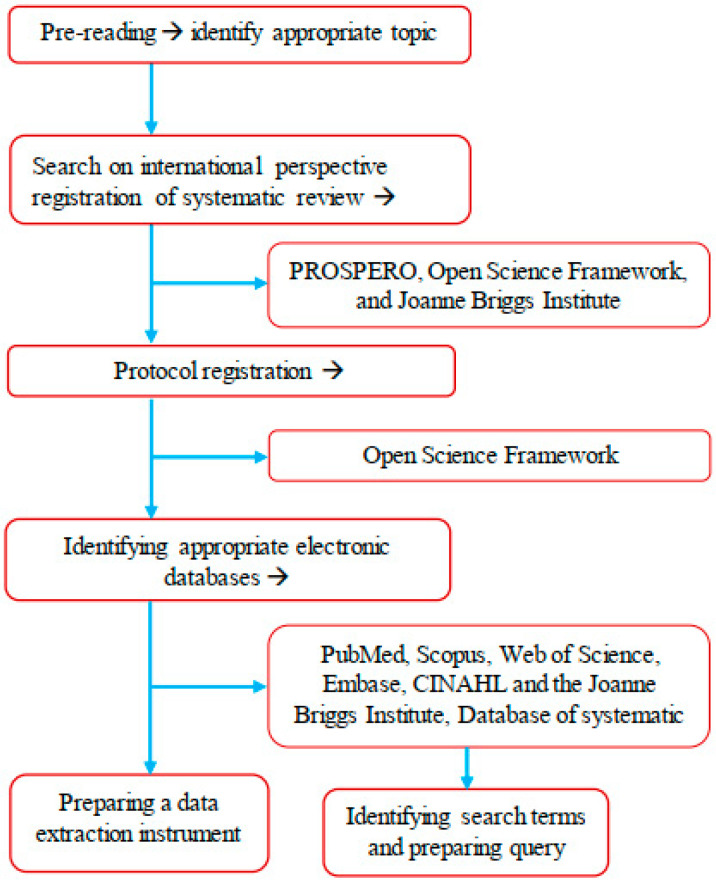
Graphic summary of the methodological strategy of the scoping review.

## Data Availability

Not applicable.

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
