# Peer review of "Association of Hypertension, Diabetes, and Cardiovascular Disease with COVID-19 in Africa: Scoping Review Protocol"

_tropicalmed, 2023, doi:10.3390/tropicalmed8060293_

Round 1

Reviewer 1 Report

-Adjust abstract to a maximum of 200 words.

-Adjust appointments to the format requested by Tropical Medicine and Infectious Disease, please check

https://www.mdpi.com/journal/tropicalmed/instructions

-Add a graphic summary of the methodological strategy of the protocol

-It would be interesting to discriminate between Type 1 or 2 Diabetes. Analyze if there are any results for this

-Line 14 African population is still scarce “are”

Grammar checking throughout the text is necessary

Author Response

Thank you very much for your comments.

Reviewer 2 Report

The topic of the future review is interesting, the protocol is well organized and promises important results. I am looking forward to see the finial manuscript, the review its self.

For this protocol I have some minor suggestions:

1.       The references should be presented in the main manuscript according to the authors' guidelines for MDPI journals.

2.       The authors could include in Introductions some other research carried out on COVID-19 patients from low to middle income countries: Tudoran, C.; Tudoran, M.; Cut, T.G.; Lazureanu, V.E.; Oancea, C.; Marinescu, A.R.; Pescariu, S.A.; Pop, G.N.; Bende, F. Evolution of Echocardiographic Abnormalities Identified in Previously Healthy Individuals Recovering from COVID-19. J. Pers. Med. 2022, 12, 46. https://doi.org/10.3390/jpm12010046

Minor English Editing is required.

Author Response

Thank you very much for your comments

Round 2

Reviewer 2 Report

The authors have answered all my questions.